# Synthesis of Ni-Cu-CNF Composite Materials via Carbon Erosion of Ni-Cu Bulk Alloys Prepared by Mechanochemical Alloying

**Sofya D. Afonnikova** [1], **Grigory B. Veselov** [1], **Yury I. Bauman** [1], **Evgeny Y. Gerasimov** [1], **Yury V. Shubin** [2], **Ilya V. Mishakov** [1,*] and **Aleksey A. Vedyagin** [1]

1   Boreskov Institute of Catalysis SB RAS, 5 Lavrentyev Ave., Novosibirsk 630090, Russia
2   Nikolaev Institute of Inorganic Chemistry SB RAS, 3 Lavrentyev Ave., Novosibirsk 630090, Russia
*   Correspondence: mishakov@catalysis.ru

**Abstract:** The unique physical and chemical properties of composite materials based on carbon nanofibers (CNFs) makes them attractive to scientists and manufacturers. One promising method to produce CNFs is catalytic chemical vapor deposition (CCVD). In the present work, a method based on carbon erosion (CE) of bulk microdispersed Ni-Cu alloys has been proposed to prepare efficient catalysts for the synthesis of CNF-based composites. The initial Ni-Cu alloys were obtained by mechanochemical alloying (MCA) of metallic powders in a planetary mill. The effect of MCA duration on the phase composition of Ni-Cu samples was studied by X-ray diffraction analysis and temperature-programmed reduction in hydrogen. It has been also revealed that, during such stages as heating, reduction, and short-term exposure to the reaction mixture ($C_2H_4/H_2/Ar$) at 550 °C, the formation of a Ni-based solid solution from the initial Ni-Cu alloys takes place. The early stages of the CE process were monitored by transmission electron microscopy combined with energy-dispersive X-Ray analysis. It was found that the composition of the catalytic particles is identical to that of the initial alloy. The morphological and structural features of the prepared Ni-Cu-CNF composites were studied by scanning and transmission electron microscopies. The textural characteristics of the composites were found to be dependent on the reaction time.

**Keywords:** nickel–copper alloys; mechanochemical alloying; carbon erosion; carbon nanofibers; CNF-based composites

## 1. Introduction

Composite materials of various natures attract a growing interest of researchers and industrialists due to their unique properties and wide range of possible applications [1–4]. In turn, carbon nanomaterials (CNMs) possess high surface area, tailored porous structure, good electrical conductivity, etc. [5–9]. Therefore, the attractiveness of CNM-based composites is also at a high level [10–18].

To prepare structured carbon nanomaterials, a method of catalytic chemical vapor deposition (CCVD) is usually applied [19–23]. This method is characterized by high efficiency, and any carbon-containing raw materials can be used as a carbon source [24–28]. CNM-based composites are widely applied in electronics [29,30], as reinforcing additives in polymer materials [31,32], and as catalysts in industrially important processes [33,34].

It should be noted that the catalytic decomposition of hydrocarbons allows for obtaining not only nanostructured carbon products but $CO_2$-free hydrogen as well [35–41]. In this regard, various catalytic compositions which were efficient in the process were developed and abundantly studied. Among such catalysts, Ni-Cu systems are of particular interest due to their low cost and high activity in the decomposition of methane and other hydrocarbons [42–47]. However, due to the high temperatures required for the decomposition of hydrocarbons, the process is complicated by the sintering of metal particles and coking

(deposition of non-structured carbon). Ultimately, it leads to the deactivation of the catalyst and a significant loss in process efficiency [48–50]. Therefore, the development of new approaches to the synthesis of active and stable Ni-Cu catalysts remains an actual problem.

In this research, the concept of carbon erosion (CE) of bulk metal precursors is utilized. This concept is based on the phenomenon of metal dusting under the action of a reactive medium [51–53]. Recently, the approach was reported to be a promising technique for the synthesis of composite materials [54,55]. During the CE process, spontaneous destruction of bulk metals or alloys in the carbon-containing atmosphere at elevated temperatures (400–600 °C) occurs. Ni and its alloys are known to be among the materials susceptible to the CE process [56,57]. The destruction of the coarse metal particles (or items) starts with the deposition of carbon at the grain boundaries [58,59]. Then, submicron metal particles capable of catalyzing the growth of nanostructured carbon are formed [58,60,61]. It is important to note that the growth of filamentous carbon such as carbon nanofibers (CNFs) or carbon nanotubes (CNTs) occurs according to the carbide cycle mechanism [62,63]. As was demonstrated recently [54], the formed M-CNF composite materials, where M is an active metal or alloy, can be considered as a self-dispersed catalyst for further use in the decomposition of hydrocarbons and the synthesis of CNF-based composites.

Another important issue is the selection of an appropriate method to prepare the metal precursor of the catalyst. In these terms, thermolysis of multicomponent precursors allows for obtaining $Ni_{1-x}M_x$ alloys of a given composition in a wide range of concentrations of alloying metal M [54,64,65]. However, this approach faces difficulty in producing large amounts of the material. Another problem is the presence of harmful by-products that need to be utilized. In this regard, the method of mechanochemical alloying (MCA) is more suitable for obtaining microdispersed alloys. MCA involves the premixing of metal powders and their subsequent alloying in a planetary mill [66,67]. The main advantage of this method is that it allows for the production of large amounts of metal precursors without any gaseous or liquid by-products.

The effectiveness of the MCA-prepared Ni-Cu alloys in the decomposition of ethylene involving the CE stage was reported previously [68,69]. It was found that the productivity of the catalyst is mainly influenced by the duration of the MCA procedure. The dispersed particles of the catalyst formed at the preparation stage are capable of catalyzing the CNF growth efficiently [70,71]. However, some questions regarding the CCVD process remained unclear. The features of CCVD of ethylene over the microdispersed Ni-Cu alloys, as well as the effects of the pretreatment conditions, have not been studied precisely yet. The purpose of the present investigation was to examine in more detail the evolution of Ni-Cu catalytic system at each stage of its generation, as well as to highlight its applicability for the targeted synthesis of nanostructured M/CNF composites.

Thereby, in the present research, the peculiarities of the formation of the Ni-Cu catalyst for the targeted synthesis of Ni-Cu-CNF composite have been studied in detail. The catalyst's precursor was prepared by the MCA method. Subsequently, it was subjected to the CE process in the ethylene atmosphere. The phase composition of the Ni-Cu precursors was studied by X-ray diffraction (XRD) analysis and temperature-programmed reduction in hydrogen (TPR-$H_2$). The Ni-Cu samples were tested in the catalytic decomposition of ethylene, and the carbon accumulation curves were analyzed. The preheating stage of the process was studied in detail by XRD and scanning and transmission electron microscopies for the first time. The energy-dispersive X-ray analysis was used to reveal the character of Ni and Cu distribution in the composition of as-formed active particles. The morphology and structural features of the Ni-Cu-CNF composites obtained via the CCVD process for 20–120 min were also examined. The textural characteristics of synthesized composites were measured by the low-temperature adsorption/desorption of nitrogen.

## 2. Materials and Methods

### 2.1. Materials

For the synthesis of Ni and Ni-Cu catalyst precursors, nickel powder (Rusredmet, St. Petersburg, Russia) and copper powder (Spetspostavka LLC, Novosibirsk, Russia) were used. For the catalytic experiments, ethylene (Nizhnekamskneftekhim, Nizhnekamsk, Russia), high-purity argon, and hydrogen were used.

### 2.2. Synthesis of Ni-Cu via Mechanochemical Alloying

A series of Ni-Cu alloys (precursors of the catalysts) were obtained by mechanochemical alloying (MCA) using an Activator 2S planetary mill (Activator LLC, Novosibirsk, Russia). Before the MCA procedure, a premix was prepared by mixing nickel and copper powders in a predefined weight ratio (Ni/Cu = 88/12). Then, 10 g of the premix was loaded into a steel jar (V = 250 mL) with stainless steel milling balls 3 mm in diameter. The weight of the balls was 340 g, which corresponded to a powder-to-balls ratio of 34 g/g.

The rotational speeds of the jars and the platform were regulated using an industrial frequency inverter VF-S15 (Toshiba, Jakarta, Indonesia). The rotational speed of the jars was 449 rpm, and the central axis speed was 956 rpm. The estimated acceleration of the milling balls was 784 m/s$^2$ (~80 g). The jars were water-cooled to avoid overheating during the MCA procedure. The activation time ($\tau$) varied in a range from 3 to 11 min. The MCA procedure was periodically stopped after 3, 5, 7, and 9 min for sampling (~100 mg). At the end of the MCA procedure, the jars were unloaded in the air. The obtained samples of Ni–Cu alloys were separated from the milling balls and weighed. A reference sample (Ni without the addition of Cu) was prepared by MCA of nickel powder (10 g) for 7 min (with intermediate stops after 3 and 5 min). The samples are designated as NiCu and Ni or NiCu(m) and Ni(m), where m is the duration of the MCA procedure.

### 2.3. Studies on the CE Process

A sample of the metal precursor (1.50 ± 0.02 mg) was placed into a basket made of foamed quartz, suspended by a quartz spring, and loaded into a flow-through quartz reactor equipped with a McBain balance (Figure 1). Then, the reactor was fed with an argon flow and heated to the reaction temperature (550 °C) with a ramping rate of 10 °C/min. Once the target temperature was reached, the sample was brought in contact with the reaction mixture containing ethylene (18 vol.%), hydrogen (59 vol.%), and argon (balance). The total flow rate of the reaction mixture was 66 L/h. Each catalytic experiment was performed for 30 min. At the end of the experiment, the reactor was cooled down to room temperature in an argon flow. The obtained carbon product was unloaded and weighed, and the specific carbon yield was calculated (g/g$_{cat}$).

### 2.4. Synthesis of the Ni-Cu-CNF Composites

The sample of the MCA-prepared Ni-Cu alloy (100.0 mg) was placed on a quartz plate, which was then installed inside a tubular reactor (Zhengzhou Brother Furnace Co., Ltd., Zhengzhou, Henan, China). Note that this reactor is characterized by a stable temperature profile along the length (±5 °C). After that, an inert flow (Ar) was purged through the reactor, and the reactor was heated up to the target temperature of 550 °C. This temperature provides stable operation of the catalysts in the ethylene decomposition reaction. Then, it was fed with the reaction mixture containing ethylene (18 vol.%), hydrogen (59 vol.%), and argon (balance). The experiments were performed for 20–120 min. After the experiment, the reactor was cooled to room temperature. The obtained carbon product was unloaded and weighed, and the values of specific carbon yield (g/g$_{cat}$) and bulk density ($\rho$) (g/L) were calculated. In order to study the evolution of the phase composition of the Ni-Cu precursor during the pretreatment stages (heating, reduction) and early stages of the CE process, the experiments were interrupted at different times. The measurement error of these experiments was ±10%.

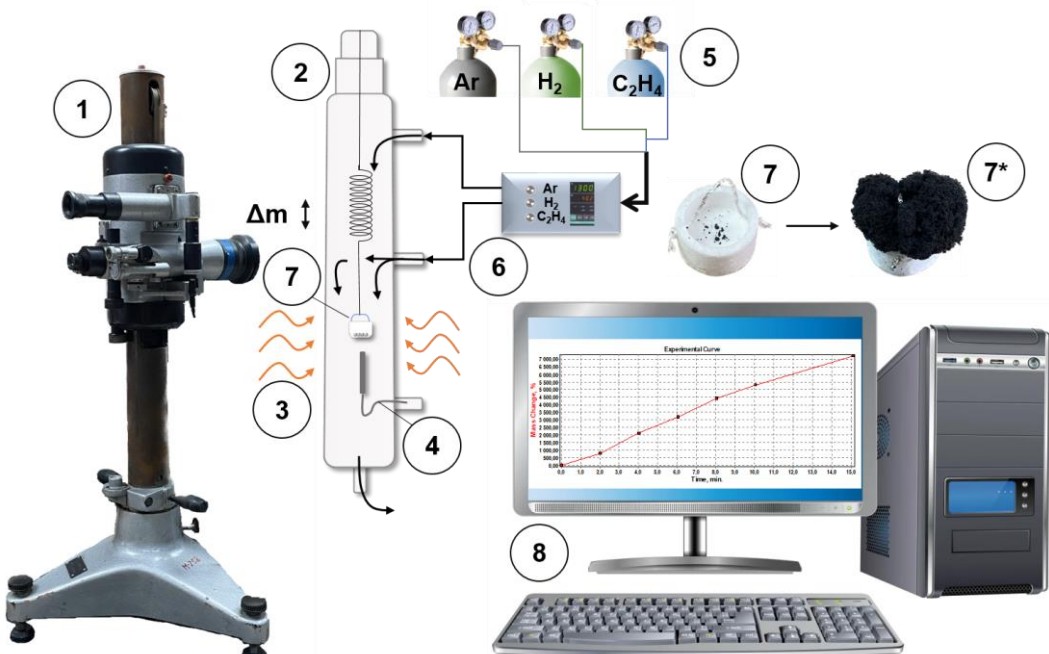

**Figure 1.** Principle scheme of the installation for catalytic experiments: 1—cathetometer; 2—flow-through quartz reactor equipped with McBain balances; 3—heating element; 4—thermocouple; 5—gaseous reagents ($C_2H_4/H_2/Ar$); 6—process (temperature and flow mass) controller; 7—a basket of foamed quartz with a loaded sample of the Ni-Cu precursor; 7*—a basket with the Ni-Cu-CNF composite formed after 30 min of reaction; 8—a personal computer with primary data of the kinetic experiment.

### 2.5. Characterization of the Ni-Cu Precursors and Ni-Cu-CNF Composites

X-ray diffraction (XRD) analysis was carried out on a Shimadzu XRD-7000 diffractometer (Shimadzu, Tokyo, Japan) with $CuK_\alpha$ radiation (Ni-filter) at a wavelength of 1.54178 Å. To determine the phase composition, scanning was performed in a 2θ range from 20° to 100° with a step of 0.05°. Phase identification was performed using the JCPDS-PDF database [72]. Accurate calculation of the cell parameters was performed by scanning in a 2θ range of 140–147° with a step of 0.05° and accumulation time in one point of 10 s. The use of the far-angle region makes it possible to minimize the error in determining the parameters of the crystal lattice. For samples with crystallite sizes ≥50 nm, the error does not exceed ±0.001 Å. Samples with small crystallite sizes (<50 nm) have broader peak profiles that lead to a decrease in the accuracy of determining the interplanar distances and, correspondingly, the lattice parameters to ±0.002–0.003 Å. The parameters of the crystal cell of solid solutions were determined by the reflection (331) position using the Powder-Cell 2.4 program [73]. The average crystallite size was calculated from the broadening of reflections (111), (200), and (220) using the Scherrer equation [74]. The crystallite size was calculated and the Pearson function was used to describe the diffraction reflections using the WinFit 1.2.1 program [75].

The reducibility of the Ni-Cu precursors was studied by a method of temperature-programmed reduction in hydrogen (TPR-$H_2$). In these experiments, a sample of the catalyst's precursor (300 mg) was loaded into a quartz reactor. Then, the reactor was heated from 30 to 700 °C in a gas mixture containing 10 vol.% $H_2$ in $N_2$ at a flow rate of 3.43 L/h. Analysis of the $H_2$ concentration at the reactor outlet was carried out using a GAMMA-100 gas analyzer (Analytpribor, Smolensk, Russia) equipped with a thermal conductivity detector. Hydrogen concentration was determined with an error of 0.01%.

The secondary structure of the MCA-prepared alloys and the morphology of the carbon product was studied by scanning electron microscopy (SEM) using a JSM-6460 (JEOL, Tokyo, Japan) instrument with magnification from 1000× to 30,000×.

The transmission electron microscopy (TEM) studies coupled with elemental mapping were performed using a Hitachi HT7700 transmission electron microscope (Hitachi Ltd., Tokyo, Japan) working at an acceleration voltage of 100 kV, with a W source, and equipped with a STEM system and a Bruker Nano XFlash 6T/60 energy-dispersive X-ray (EDX) spectrometer (Bruker Nano GmbH, Berlin, Germany). Before examination by TEM, the sample of Ni-Cu/CNF composite was suspended in ethanol and then deposited over the TEM grid coated with a perforated carbon film.

The textural characteristics of the composite materials were determined by low-temperature nitrogen adsorption/desorption. Isotherms were measured at 77 K on an automated ASAP-2400 instrument (Micromeritics, Norcross, GA, USA). The samples were degassed at 300 °C for 6 h prior to the measurements. The specific surface area (SSA) was calculated via the Brunauer–Emmett–Teller (BET) method. The pore volume ($V_p$) was determined at a relative pressure $P/P_0 = 0.99$ and corresponded to pores <200 nm in size. The micropore volume ($V_\mu$) was calculated using the $\alpha_s$ method [76]. The measurement error was 25% for Ni-Cu samples and 5% for Ni-Cu-CNF composites.

## 3. Results and Discussion

### 3.1. Preparation of the Ni-Cu Precursors and Ni-Cu-CNF Composites

Figure 2 shows the scheme of the MCA method chosen for the preparation of Ni-Cu precursors. The presented SEM data correspond to the NiCu samples obtained at different MCA times ($\tau$). As known, the MCA method is based on the intensive mechanical action of milling balls on a mixture of metal powders [67]. During the process, metal particles interact closely with each other, gradually forming an alloy. As was recently reported [69], at the first stages of the MCA procedure, plastic deformation of particles occurs, leading to the formation of layered plates. Then, these plates crumble and form a large number of small fragments of the alloy, which undergo secondary agglomeration.

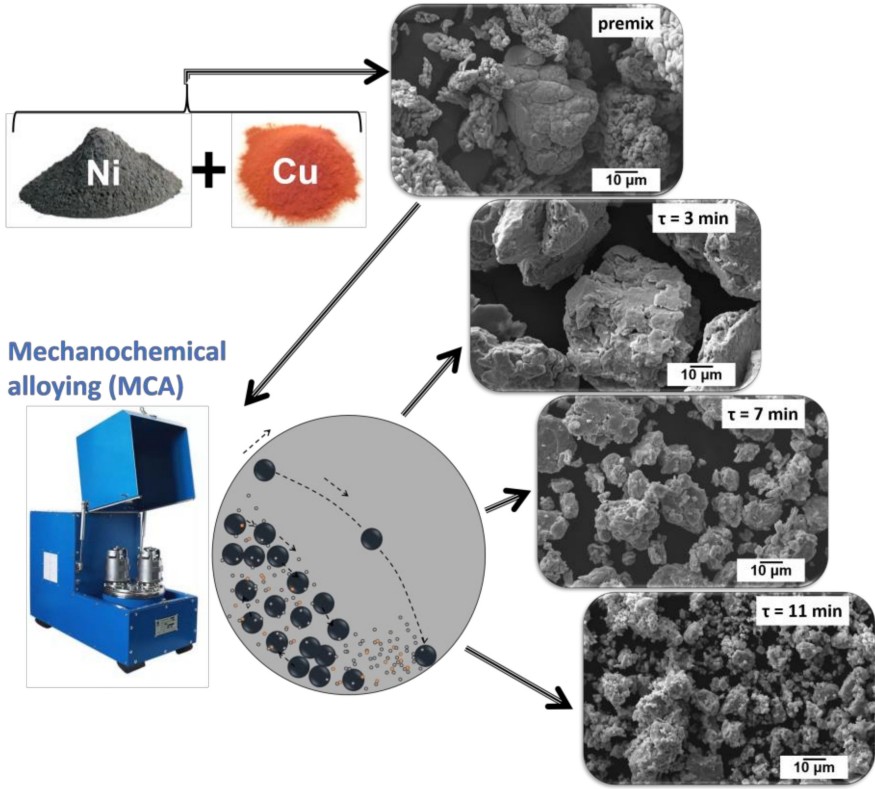

**Figure 2.** Principle scheme of the synthesis of Ni-Cu metal precursor via MCA using an Activator-2S planetary mill. The presented SEM images correspond to the premix sample and the samples after MCA during 3, 7, and 11 min.

The premix sample and the samples after MCA were exposed to contact with the reaction mixture containing ethylene in the reactor equipped with a McBain balance. All experiments were performed at temperature of 550 °C, which was previously shown to be appropriate for the catalytic pyrolysis of ethylene [69]. The weight of the samples increased during the experiment due to the formation of the nanostructured carbon product. The carbon accumulation curves are shown in Figure 3. Curves for the premix sample and pure nickel (after MCA for 7 min) are presented for comparison. Table 1 summarizes the carbon yield values for all studied samples. By comparing the data for the Ni(7) and NiCu(7) samples, it can be concluded that the addition of Cu to Ni increases the catalytic activity of the system noticeably (Figure 3a). Thus, the carbon yield increases by 2.5 times from 50.7 to 129.9 g/$g_{cat}$ (Table 1). Such an effect is naturally expected, since copper is often used as an agent stabilizing the catalytic activity of nickel [45,70].

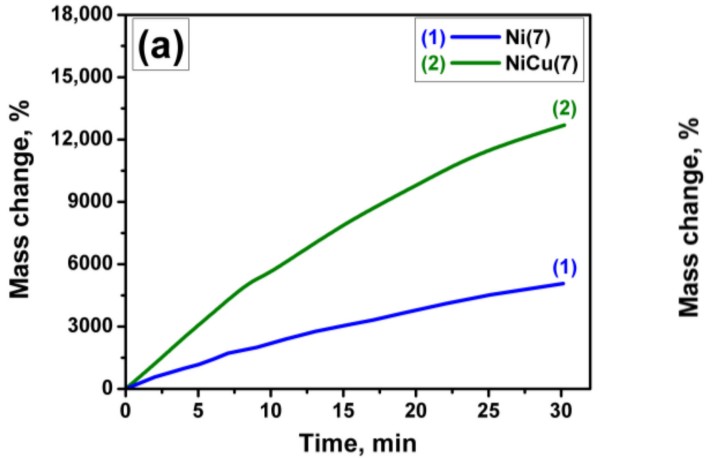 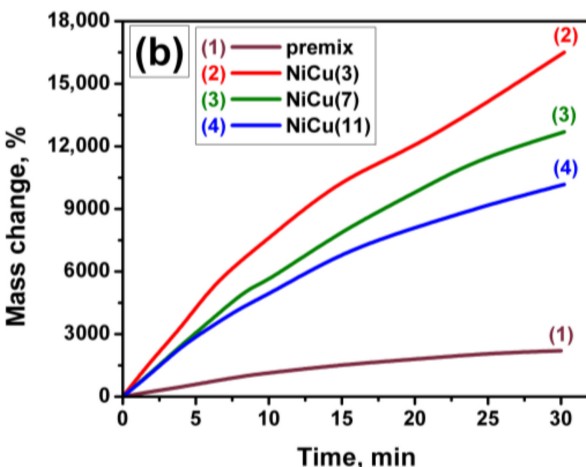

**Figure 3.** Accumulation of carbon (CNF) during the decomposition of the $C_2H_4/H_2/Ar$ reaction mixture over the samples at 550 °C: (**a**) comparison of pure Ni(7) and NiCu(7) alloy; (**b**) effect of the MCA time.

**Table 1.** Carbon yield values, XRD characteristics, and TPR-$H_2$ data for the samples under study.

| Sample | Activation Time ($\tau$, min) | Carbon Yield, g/$g_{cat}$ | Lattice Parameter, Å | Crystallite Size, nm | Hydrogen Uptake, mmol/g | | SSA, m²/g |
|---|---|---|---|---|---|---|---|
| | | | | | Peak A | Peak B | |
| Premix | - | 22 ± 2 | 3.525 ± 0.001 | 50 ± 10 | 0.17 ± 0.01 | - | 0.5 ± 0.3 |
| Ni(7) | 7 | 51 ± 5 | 3.525 ± 0.001 | 11 ± 3 | 0.77 ± 0.02 | −0.12 ± 0.02 | 1.7 ± 0.4 |
| NiCu(3) | 3 | 165 ± 17 | 3.523 ± 0.002 | 17 ± 5 | 0.18 ± 0.01 | −0.14 ± 0.02 | 0.4 ± 0.1 |
| NiCu(7) | 7 | 127 ± 13 | 3.532 ± 0.003 | 9 ± 3 | 0.76 ± 0.01 | −0.24 ± 0.02 | 1.7 ± 0.4 |
| NiCu(11) | 11 | 102 ± 10 | 3.535 ± 0.003 | 8 ± 3 | 1.20 ± 0.01 | −0.21 ± 0.02 | 1.9 ± 0.5 |

As seen in Figure 3b, the duration of the MCA procedure also affects the CCVD process noticeably. The short-term alloying (3 min) of the Ni and Cu premix leads to a significant increase in their productivity in the catalytic decomposition of ethylene. The maximum yield of 165.1 g/$g_{cat}$ was recorded in this case, which is 3.8 times higher than that of the premix sample (Table 1). Recently, a similar trend was reported for the Ni-Cu system prepared by the same procedure but using milling balls of a larger diameter (5 mm) [68]. However, the optimal MCA time corresponding to the highest carbon yield was 5 min in that case.

It is worth noting that all the carbon accumulation curves presented in Figure 3 are of the same character. During the first minutes of the reaction, the curve has a smooth slope. This indicates the absence of an induction period of the CCVD process for these samples. Both the processes, disintegration of the alloy and formation of the carbon product, occur

simultaneously. For the next ~20 min, an intensive deposition of nanostructured carbon is observed. Finally, the apparent carbon accumulation rate decreases for all the samples, which can be reasoned by the steric (or volumetric) factor. Thus, at high values of carbon yield, the deposited carbon could not fit into the reactor volume. It clings to the walls of the reactor, which affects the McBain spring-stretching process and mispresents the measured values. For evidence, the quartz basket with the deposited carbon product is shown in Figure 1. Thereby, it can be concluded here that the best catalytic performance is exhibited by the Ni-Cu system subjected to the MCA procedure for a short time.

At the next step of the research, the phase compositions of the selected Ni-Cu samples were examined. According to XRD data, NiO is registered for the Ni-Cu samples after a prolonged MCA procedure (more than 7 min). Note the detailed XRD of similar Ni-Cu samples was reported previously [69].

In order to estimate the amount of CuO and NiO in the composition of the Ni-Cu samples, the TPR-H$_2$ technique was carried out. First, two reference samples of bulk nickel and copper oxides were studied. These samples were prepared by the decomposition of corresponding nitrates at 500 °C. The TPR-H$_2$ profiles of these samples are compared in Figure 4a. As is seen, CuO reduces in a temperature range of 200–350 °C with a maximum hydrogen uptake at 310 °C. The reduction of NiO takes place at higher temperatures, in a range of 300–450 °C, with a maximum at 410 °C. In both cases, the hydrogen uptake peaks are asymmetrical. The appearance of a left shoulder indicates the presence of dispersed surface species of these oxides.

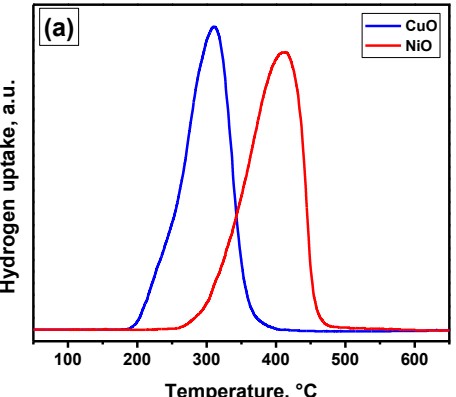 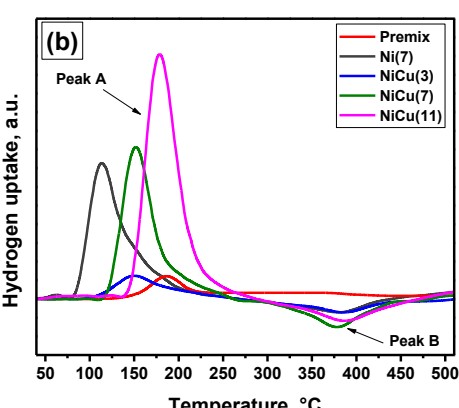

**Figure 4.** TPR-H$_2$ profiles for the studied samples: (**a**) comparison of bulk CuO and NiO oxides; (**b**) comparison of the Ni and NiCu samples under study.

The TPR-H$_2$ profiles of the Ni-Cu samples subjected to the MCA procedure for different times are presented in Figure 4b. The calculated hydrogen uptake values are given in Table 1. It can be seen that there is only one peak in the profile of the premix sample with a maximum at 185 °C (peak A), which appeared at a lower temperature with respect to the reference samples (Figure 4a). It is worth noting that the simultaneous presence of CuO and NiO oxides in the sample can simplify their reduction. For instance, the reduction of the bimetallic Ni-Cu/SiO$_2$ system was reported to begin at lower temperatures if compared with monometallic Ni/SiO$_2$ and Cu/SiO$_2$ samples [77]. This can be explained by the ability of hydrogen to adsorb on the surface of metallic copper, which is formed by the reduction of copper oxide. This adsorbed hydrogen simplifies the reduction of NiO species due to the spillover effect. In addition, as is shown for the reduced/reoxidized Ni-Cu/ZnO samples [78], CuO and NiO oxides reduced simultaneously, i.e., just one peak corresponding to the reduction of both oxides was observed. In the case of the Ni-Cu premix (Figure 4b), hydrogen uptake is mainly associated with the reduction of the thin film of the surface oxides, and most of the metals are in the metallic state (Ni$^0$ and Cu$^0$), which ensures the reduction of oxides at low temperatures.

The MCA treatment of the premix for 3 min leads to some changes in the TPR-$H_2$ profile. First, the oxide reduction peak A shifted to the lower-temperature region. At the same time, the hydrogen uptake value increased significantly. Second, a negative peak B with an extremum at 385 °C has appeared. Peak B can be associated with hydrogen desorption. Znak and Zieliński [79,80] have reported that desorption of hydrogen from the surface of bulk Ni can take place in several stages with maxima at 227 and 277 °C. In our case, the temperature of hydrogen desorption is slightly higher, which may be due to the experimental conditions. In our experiments, the reaction mixture contains just 10% hydrogen (~76 mmHg), which can slightly shift the reaction equilibrium toward adsorption. In general, the adsorption of hydrogen can occur at low temperatures or simultaneously with the reduction of oxides (peak A). Another possible explanation is that hydrogen could be trapped in point defects and dislocations as well as at grain boundaries [81,82]. It should be emphasized that the appearance of such defects is a character of the MCA procedures [67].

After 7 min of the MCA treatment, the intensities of both peaks A and B are increased. It is worth noting that reflections corresponding to the NiO phase are present in the XRD patterns. By comparing two samples MCA-treated for the same time, Ni(7) and NiCu(7), it is seen that, in the absence of copper, peak A is shifted to the low-temperature region (113 °C), although its area remains almost the same. At the same time, the position of peak B remained unchanged, while its area decreased twofold (Table 1). It seems that the addition of copper does not contribute to an increase in the content of oxides but leads to an increase in the number of sites capable of adsorbing hydrogen.

A further increase in the MCA time to 11 min increases the intensity of peak A and results in its displacement to the high-temperature region. As shown previously [69], the size of the Ni-Cu alloy particles increases at low MCA times and begins to decrease at an MCA time of 3 min and above. As a result, the available surface of the alloy also grows (Table 1), which leads to a higher content of surface oxides. Thus, the amount of adsorbed hydrogen increases as well. It is natural to assume that the alloy samples subjected to the MCA procedure for a longer time possess higher defectiveness. Consequently, the longer the MCA procedure is, the higher the specific surface area of the particles is, and the higher the amount of adsorbed hydrogen is.

Another important question concerning the duration of the MCA procedure is about the minimal time required for the formation of Ni-Cu alloy. Previously, it was supposed that a short time of MCA treatment is not enough to form the alloy. On the other hand, the NiCu(3) sample demonstrated the best productivity of 165.1 g/g$_{cat}$ (Table 1), whereas the premix sample showed just 22.2 g/g$_{cat}$. Therefore, the effects of different treatment conditions on the phase composition of the catalyst's precursor were investigated in order to reveal the reasons for such a high activity of the NiCu(3) sample. It should be noted that not only the MCA activation could affect the state of the catalyst. Before the reaction, the catalyst's precursor should be heated to the reaction temperature of 550 °C in an inert medium. In this regard, it was interesting to track the evolution of the phase composition of the sample starting from the premix state. The results of the XRD study are shown in Figure 5. Numerical XRD data (lattice parameters and crystallite size) for the studied samples are summarized in Table 2.

By comparing the XRD patterns of the premix sample (pattern 1) with the NiCu(3) sample (pattern 2), it can be concluded that the MCA treatment of the premix for 3 min does not lead to the formation of the alloy. The reflections related to the metallic Cu phase are still present in the pattern. The same conclusion follows from the matching of the lattice parameter for the NiCu(3) sample (a = 3.523 (2) Å) with that of pure nickel (a = 3.5238 Å). Further heating of the sample in Ar up to 550 °C results in the alloy formation accompanied by the disappearance of the reflections related to metallic Cu (pattern 3). Additional reduction of the preheated sample in the $H_2$/Ar mixture for 1 min does not cause any changes in the phase composition (pattern 4). Thereby, the short-term mechanochemical alloying (3 min) of the Ni-Cu premix does not lead to the appearance of the alloy during

the procedure but facilitates alloy formation during the heating of the MCA-treated sample to the reaction temperature. It is important to note that the reaction temperature of 550 °C is significantly lower than the melting points of Cu (1085 °C) and Ni (1455 °C). Finally, the short-time exposure of the preheated sample to the reaction mixture results in the appearance of a new reflection near 2θ = 26° (pattern 5). This reflection can be attributed to the carbon phase. All other reflections in this pattern correspond to the *fcc* lattice of Ni (Table 2).

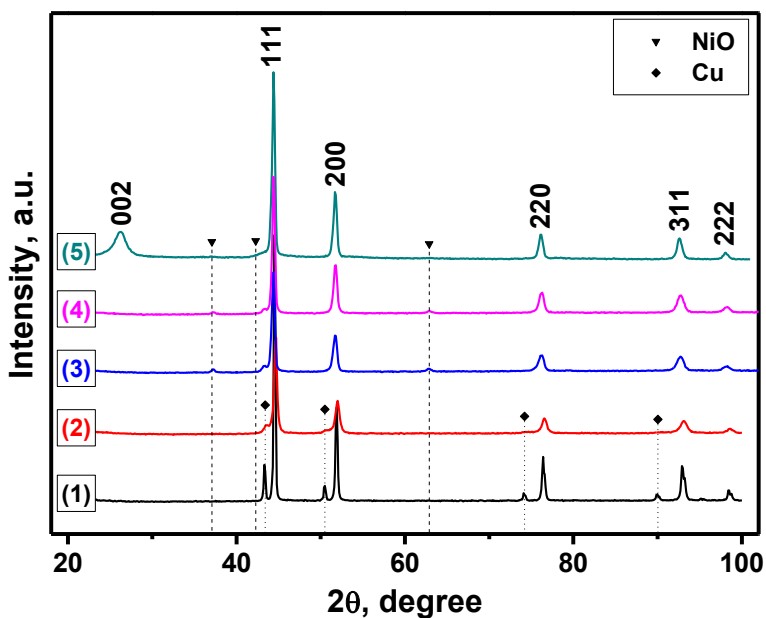

**Figure 5.** XRD patterns of the Ni-Cu samples: (1) premix; (2) after MCA for 3 min; (3) after MCA for 3 min and heating in Ar to 550 °C; (4) after MCA for 3 min, heating in Ar to 550 °C, and maintaining in $H_2$/Ar for 1 min; (5) after MCA for 3 min, heating in Ar to 550 °C, and exposure to the reaction mixture ($C_2H_4$/$H_2$/Ar) for 1 min.

**Table 2.** XRD data for the Ni-Cu samples collected at different stages of the study.

| Sample | Reaction Conditions | Lattice Parameter, Å | Crystallite Size, nm | Phase Composition |
|---|---|---|---|---|
| Premix | - | 3.525 ± 0.001 | 52 ± 11 | A mixture of Ni and Cu phases |
| NiCu(3) | MCA for 3 min | 3.523 ± 0.002 | 17 ± 5 | A mixture of Ni and Cu phases |
| NiCu(3)550 | MCA for 3 min; heating in Ar to 550 °C | 3.535 ± 0.002 | 16 ± 5 | Ni-Cu alloy |
| NiCu(3)550/1′ | MCA for 3 min; heating in Ar to 550 °C; reduction with $H_2$ for 1 min at 550 °C | 3.532 ± 0.001 | 18 ± 5 | Ni-Cu alloy |
| NiCu(3)550/1″ * | MCA for 3 min; heating in Ar to 550 °C; exposure to the reaction mixture for 1 min at 550 °C | 3.535 ± 0.001 | 21 ± 6 | Ni-Cu alloy |

* The sample contains 80 wt.% of carbon.

The obtained NiCu(3)550/1″ sample was examined in detail by SEM and TEM (Figures 6 and 7). As is seen, the material is represented by rather short carbon filaments (nanofibers) grown on the particles of the Ni-Cu alloy (Figure 6a,b). It is worth noting that the formed active metal particles are mainly of biconical shape (Figure 6c). The carbon filaments grow predominantly in two opposite directions. As follows from the microscopic data, 1 min of exposure to the reaction mixture is enough for the almost complete destruction of the bulk alloy and the formation of active particles catalyzing

the growth of CNF (Figure 6b,c). This confirms the previously mentioned observation that, in the case of dispersed Ni-Cu alloys, the CE process proceeds without an induction period. As usual, a prolonged induction period is required for the bulk metal items to be disintegrated. However, some alloy particles, which are still being disintegrated, are also observed (Figure 6d). It is worth noting that some other catalytic particles undergo a secondary disintegration process within 1 min of the reaction. This is evidenced by the presence of thin filaments that form tangles around the catalytic particle (Figure 6b). Similar tangles of thin carbon nanofibers were obtained previously when decomposing the mixture of $C_2$–$C_4$ hydrocarbons on the Ni-Cr alloy [83]. However, in that case, the secondary disintegration process was initiated after the pre-synthesis of the self-organizing catalyst in the 1,2-dichloroethane medium for 1 h.

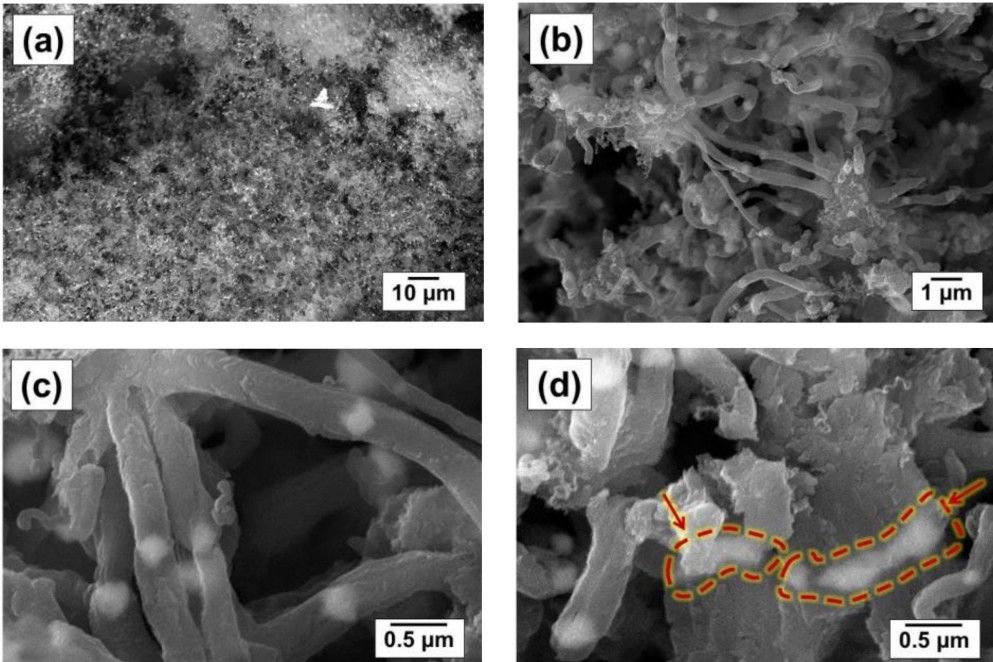

**Figure 6.** SEM images (backscattered electron mode) of the Ni-Cu-CNF composite obtained via the decomposition of ethylene over the NiCu(3) sample at 550 °C for 1 min: (**a**) magnification 1000×; (**b**) magnification 10,000×; (**c,d**) magnification 30,000×. Arrows show the particles still undergoing the disintegration process.

Since the Ni-Cu alloy was formed during the heating of the NiCu(3) sample to the reaction temperature, the elemental composition of the obtained particles was analyzed in detail by TEM (Figure 7). According to the EDX mapping, the Ni and Cu atoms are predominantly located in the same areas related to the active particles. The EDX data also suggest that no redistribution of the Ni and Cu atoms occurs during the spontaneous formation of catalytic particles.

For a single Ni-Cu particle (Figure 8), EDX mapping showed that, throughout the particle length, the distribution of Ni and Cu atoms is practically uniform; this agrees with the data on the homogeneity of Ni-Cu particles according to XRD data (Figure 5, Table 2). By comparing the obtained results with the literature data, it can be stated that the decomposition of ethylene is not accompanied by the etching of the alloying element from particles [54,55], unlike when it was observed in the case of chlorine-substituted hydrocarbons [84,85].

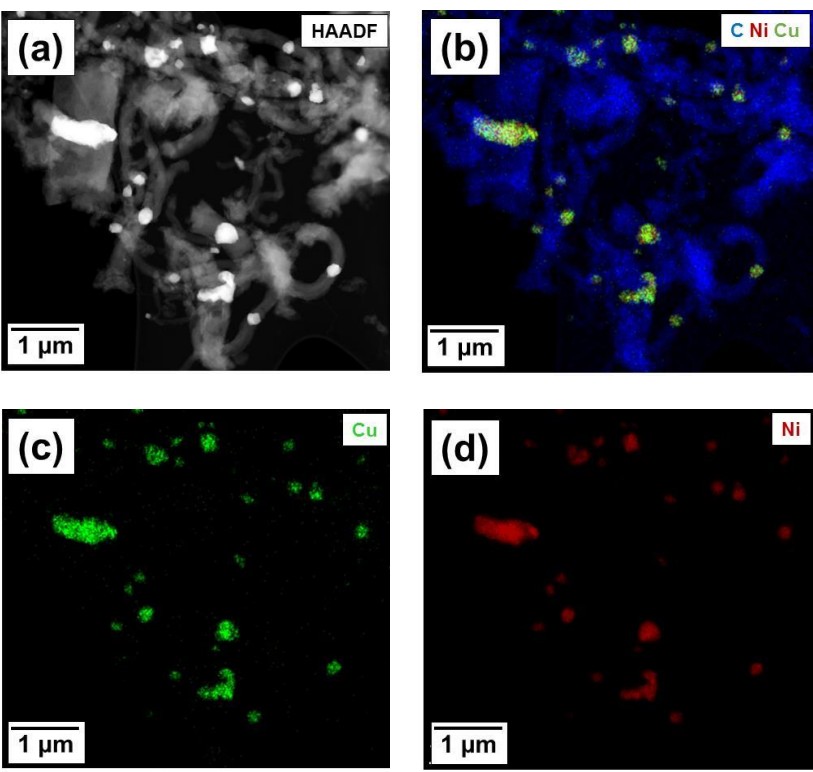

**Figure 7.** Data of EDX mapping for the Ni-Cu-CNF composite obtained via the decomposition of ethylene over the NiCu(3) sample at 550 °C for 1 min: (**a**) high-angle annular dark field; (**b**) distribution of carbon, nickel, and copper; (**c**) distribution of copper; (**d**) distribution of nickel.

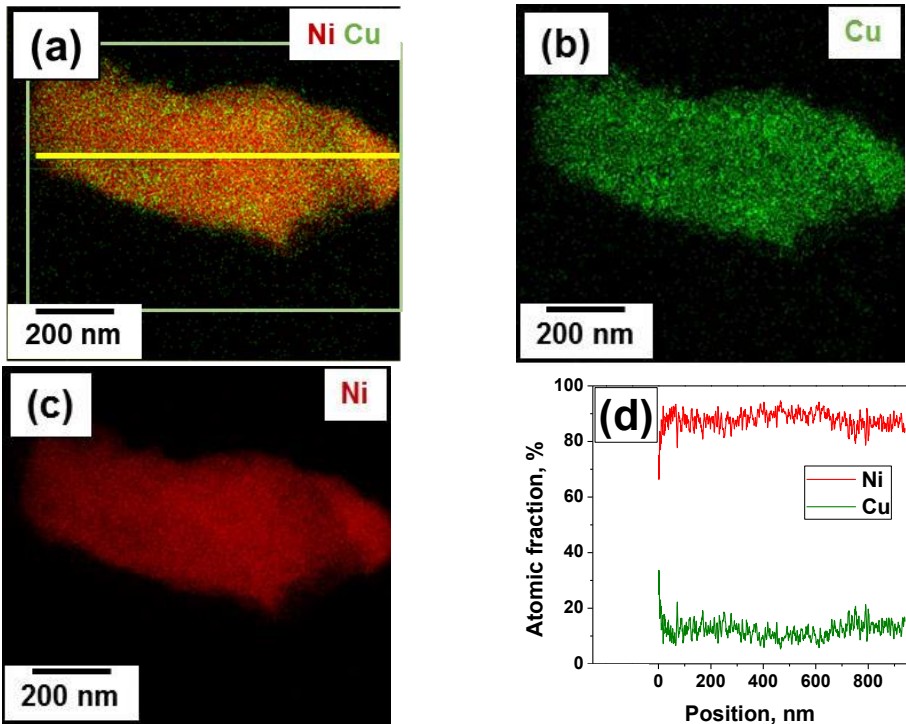

**Figure 8.** Data of EDX mapping for the single Ni-Cu particle of the Ni-Cu-CNF composite obtained via the decomposition of ethylene over the NiCu(3) sample at 550 °C for 1 min: (**a**) distribution of carbon, nickel, and copper; (**b**) distribution of copper; (**c**) distribution of nickel; (**d**) atomic fraction of the elements along the particle length (shown by the yellow line).

### 3.2. Characterization of the Ni-Cu-CNF Composites

At the final stage of the research, the Ni-Cu-CNF composites prepared at different reaction times were characterized by electron microscopy and low-temperature nitrogen adsorption. The representative amounts of the composite samples were synthesized in a tubular quartz reactor where 100 mg of the Ni-Cu precursor was loaded. The values of carbon yield ($Y_C$) are presented in Table 3. As is seen, an extension of the reaction time from 20 to 120 min leads to an increase in the carbon yield from 32 to 148 g/g$_{cat}$. It is worth noting that this approach provides the possibility to control the concentration of metal particles in the final composition of the composite.

**Table 3.** Characteristics of the Ni-Cu-CNF composites obtained via the decomposition of ethylene over the NiCu(3) sample at 550 °C at various times.

| Sample | $t_r$, min | $Y_C$, g/g$_{cat}$ | $C_{Me}$, wt.% | $\rho$, g/L | SSA, m$^2$/g | $V_p$, cm$^3$/g | $V_\mu$, cm$^3$/g | $D_{avg}$ (4$V_p$/SSA), nm |
|---|---|---|---|---|---|---|---|---|
| Ni-Cu-CNF_20 | 20 | 32 ± 3 | 3.1 ± 0.3 | 25 ± 3 | 159 ± 8 | 0.15 ± 0.01 | 0.040 ± 0.002 | 3.7 ± 0.2 |
| Ni-Cu-CNF_40 | 40 | 62 ± 6 | 1.6 ± 0.2 | 25 ± 3 | 158 ± 8 | 0.15 ± 0.01 | 0.042 ± 0.001 | 3.8 ± 0.2 |
| Ni-Cu-CNF_60 | 60 | 90 ± 9 | 1.1 ± 0.1 | 25 ± 3 | 158 ± 8 | 0.16 ± 0.01 | 0.041 ± 0.001 | 3.9 ± 0.2 |
| Ni-Cu-CNF_90 | 90 | 120 ± 12 | 0.8 ± 0.1 | 25 ± 3 | 134 ± 7 | 0.15 ± 0.01 | 0.035 ± 0.002 | 4.4 ± 0.2 |
| Ni-Cu-CNF_120 | 120 | 148 ± 15 | 0.7 ± 0.1 | 30 ± 3 | 69 ± 4 | 0.09 ± 0.01 | 0.014 ± 0.001 | 5.3 ± 0.3 |

SEM images of the Ni-Cu-CNF composites obtained via the decomposition of ethylene over the NiCu(3) sample at 550 °C for 20–120 min are presented in Figure 9. The composite material is represented by carbon filaments with catalytic particles embedded in their structure. Such carbon composite materials with embedded metal particles were reported to exhibit high catalytic activity in electrocatalytic reactions [55]. SEM data confirm the complete disintegration of the alloy in all cases. It was found that the prolongation of the reaction time decreases the average CNF diameter from 500 to 300 nm (Figure 9f). It is worth noting that the CNF thickness changes insignificantly over time. It should be noted that the filaments with a rough, uneven surface are observed in each case. The number and the length of the thin fibers with a diameter below 150 nm increase significantly with time. As is seen, such filaments are formed by the secondary disintegration of the active particles.

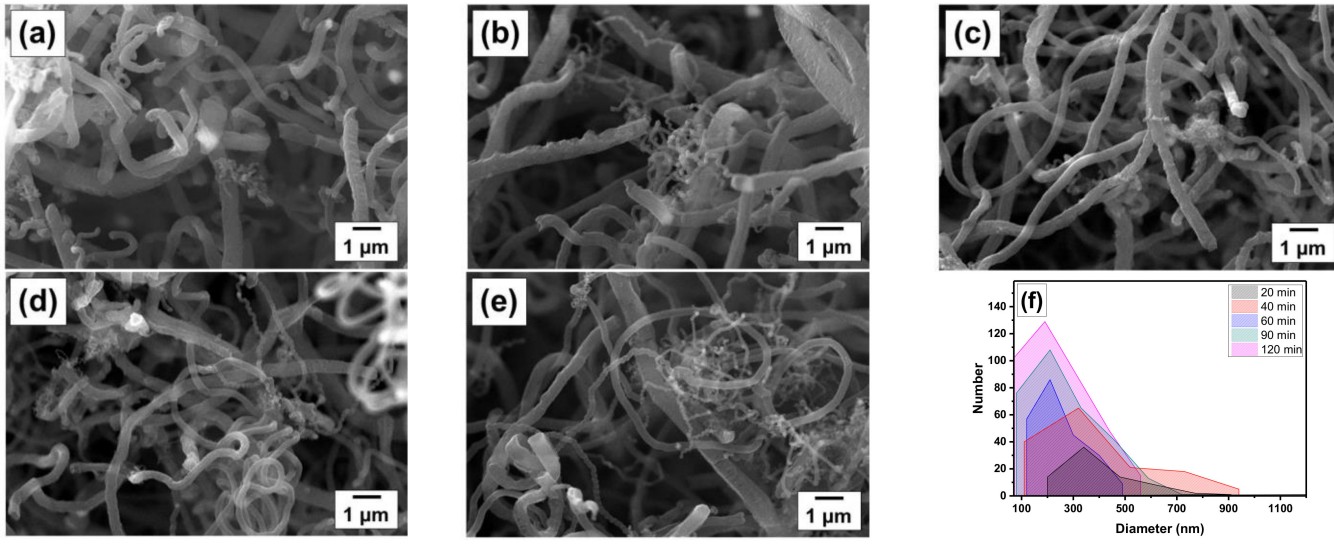

**Figure 9.** SEM images of the Ni-Cu-CNF composites obtained via the decomposition of ethylene over the NiCu(3) sample at 550 °C for: (**a**) 20 min; (**b**) 40 min; (**c**) 60 min; (**d**) 90 min; (**e**) 120 min. Corresponding distributions of the CNF diameters (**f**).

An example of the catalytic particle formed during the CE process is shown in Figure 10a. It can be seen that carbon filaments grow from this particle in three direc-

tions. A closer look at the structure of nanofibers shows that the carbon layers in these conditions have a stacked type of packing (Figure 10b). At the same time, the fibers have a loose and rough structure around the edges. However, the high-resolution TEM images of the filament suggest that the carbon layers are quite tightly stacked together. In addition, Figure 10c shows that the carbon layers at the ends are not separate but are connected with each other by bridges.

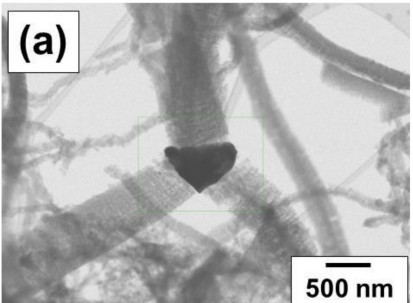 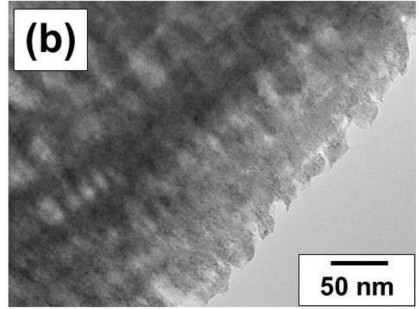 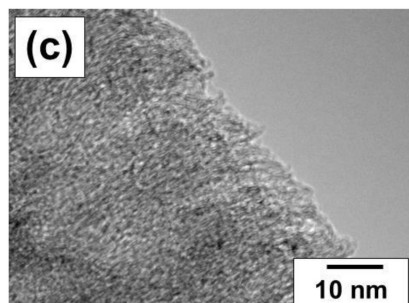

**Figure 10.** TEM images of the Ni-Cu-CNF_40 composite obtained via the decomposition of ethylene over the NiCu(3) sample at 550 °C for 40 min: (**a**) magnification 10,000×; (**b**) magnification 100,000×; (**c**) magnification 500,000×.

Figure 11 presents nitrogen adsorption/desorption isotherms recorded for the Ni-Cu-CNF_20 and Ni-Cu-CNF_120 samples. The isotherms have similar shapes that correspond to Type II according to the IUPAC classification [86]. In addition, both isotherms possess a hysteresis loop, which can be assigned to either Type H3 or H4. It can be seen that the adsorption and desorption branches of the isotherms are not connected even at low values of relative pressure. A similar phenomenon was observed previously by Maksimova et al. [87]. The non-equilibrium character of the desorption branch was attributed to the swelling of the carbon material.

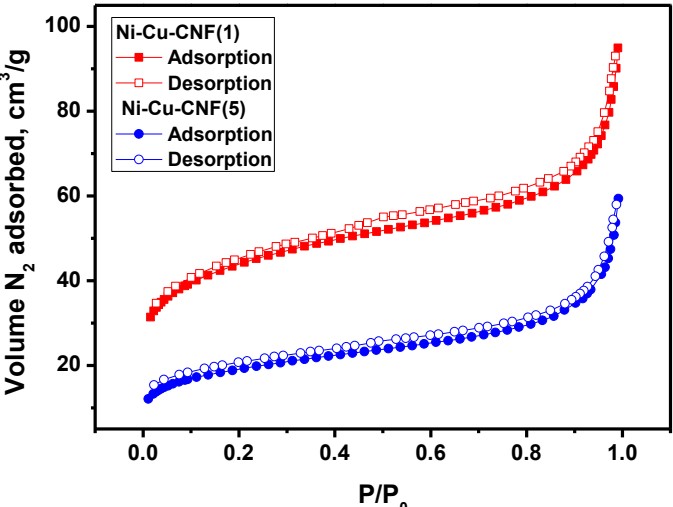

**Figure 11.** Low-temperature nitrogen adsorption/desorption data for the Ni-Cu-CNF_20 and Ni-Cu-CNF_120 composites obtained via the decomposition of ethylene over the NiCu(3) sample at 550 °C for 20 and 120 min, respectively.

The textural parameters calculated from the isotherms are shown in Table 3. It is clear that the composites obtained via the decomposition of ethylene for 20–60 min are very similar in texture. The calculated $4V_p/SSA$ values testify to the presence of mesopores in the composites. On the other hand, the presence of both meso- and micropores can be attributed to the roughness of the CNF surface [88], which is evident from the TEM images (Figure 10b,c). An increase in the duration of the CCVD process from 60 to 120 min

results in a noticeable decrease in SSA and pore volume. These data are consistent with the bulk density measurements. Thus, the observed differences in texture can be assigned to two factors:

1. The diameter and morphology of the growing fibers can be changed over time due to the changeable dispersion of metal particles (primary and secondary disintegration).
2. The surface of the already-formed CNF material can undergo restructuring due to its interaction with the gaseous hydrogen, which leads to the formation of $CH_4$ and changes in the surface structure.

According to the SEM data discussed above, the morphology of the carbon fibers does not depend on the reaction time strongly. If anything, secondary disintegration of the metal particles leads to the formation of thinner CNF with higher SSA. By excluding the second factor from the consideration, it is evident that, within the 148 g/$g_{cat}$ of the Ni-Cu-CNF_120 sample, at least 90 g/$g_{cat}$ (61 wt.%) of CNF should correspond to the structural type similar to that of Ni-Cu-CNF_60 with the SSA of 158 m$^2$/g. In this, the SSA of the Ni-Cu-CNF_120 sample should be at least 96 m$^2$/g. These simple calculations confirm that the restructuring of the CNF surface due to its interaction with hydrogen indeed takes place. This is, in fact, the main factor affecting the textural characteristics. As a result, the loss of microporosity and widening of mesopores takes place.

Nevertheless, it is worth noting that the obtained Ni-Cu-CNF samples are characterized by a high specific surface area (~160 m$^2$/g) and a low bulk density (25–30 g/L). Such a combination of the textural parameters makes these materials sufficiently promising for further use as composites in catalytic and electrocatalytic applications.

## 4. Conclusions

In the present work, a series of Ni-Cu alloys were synthesized by the mechanochemical alloying method. To prepare the Ni-Cu-CNF composites, catalytic chemical vapor deposition of ethylene was used. The duration of the MCA procedure was shown to significantly affect the activity of the Ni-Cu system. Thus, MCA treatment of the Ni-Cu premix for just 3 min increases the carbon yield in the 30 min CCVD process from 22.2 to 165.1 g/$g_{cat}$. According to TPR-H$_2$, all MCA-treated Ni-Cu alloys are characterized by the presence of a peak at temperatures of 120–200 °C, which corresponds to the reduction of a surface oxide film. Another negative peak is observed at T = 385 °C. This peak can be attributed to the desorption of hydrogen trapped in point defects and dislocations of the Ni-Cu alloy. Based on XRD data, the MCA treatment of the Ni-Cu premix for 3 min does not result in the alloy formation. It is represented by a mixture of metals. The formation of an alloy occurs during the heating of this sample in an inert atmosphere to the reaction temperature (550 °C). According to scanning and transmission electron microscopies, during the carbon erosion process, the initial microdispersed Ni-Cu alloy disintegrates on small metal particles, which catalyzes the growth of carbon nanofibers. EDX mapping confirmed that the distribution of metal atoms in these particles is uniform and corresponds to the Ni-Cu weight ratio of ~88/12, which was specified during the synthesis. Finally, a series of Ni-Cu-CNF composites was fabricated via the CCVD process in a scaled-up tubular quartz reactor. The duration of the experiments varied in range from 20 to 120 min. The obtained composites, where Ni-Cu particles are embedded in the structure of carbon nanofibers, are characterized by a high specific surface area (~160 m$^2$/g) and a low bulk density (25–30 g/L). Such materials seem to be promising for further application in catalysis and electrocatalysis.

**Author Contributions:** Conceptualization, I.V.M., S.D.A. and A.A.V.; methodology, S.D.A., Y.I.B., G.B.V. and Y.V.S.; investigation, S.D.A., G.B.V., Y.I.B., Y.V.S. and E.Y.G.; writing—original draft preparation, S.D.A. and G.B.V.; writing—review and editing, I.V.M., Y.V.S. and A.A.V.; visualization, S.D.A., G.B.V. and E.Y.G.; supervision, Y.V.S., A.A.V. and I.V.M.; funding acquisition, I.V.M. All authors have read and agreed to the published version of the manuscript.

**Funding:** This work was supported by the Ministry of Science and Higher Education of the Russian Federation within the governmental order for Boreskov Institute of Catalysis (project AAAA-A21-121011390054-1). The physicochemical characterization of samples was supported by the Russian Science Foundation (project No. 22-13-00406, https://rscf.ru/en/project/22-13-00406, accessed on 30 March 2023, BIC SB RAS).

**Data Availability Statement:** Data are contained within the article.

**Acknowledgments:** Analysis of the physicochemical properties of the samples was performed using the equipment of the "National Center for Catalyst Research". The authors are grateful to A.N. Serkova and M.N. Volochaev for help in the microscopic analysis of Ni-Cu-CNF samples (SEM, TEM), as well as D.M. Shivtsov and A.B. Ayupov for help with nitrogen adsorption/desorption measurements.

**Conflicts of Interest:** The authors declare no conflict of interest. The funders had no role in the design of the study; in the collection, analyses, or interpretation of data; in the writing of the manuscript; or in the decision to publish the results.

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
