# Peer review of "Synthesis of Ni-Cu-CNF Composite Materials via Carbon Erosion of Ni-Cu Bulk Alloys Prepared by Mechanochemical Alloying"

_jcs, doi:10.3390/jcs7060238_

Round 1

Reviewer 1 Report

Dear author and editor. I find the paper interesting, relevant and well performed and should be published. I have a few comments that I think may be relevant:

The quality/resolution of figures 1 and 2 needs to be improved.

The text in figure 3 is too small for old eyes.

On page 6 it is claimed that the slopes in figure 3 are all 45 degrees. It does not look like that in the figure. Please comment or revise. 

On page 11 it says that the atomic % ratios are homogeneous and near the target value. Figure 8d clearly disagrees with this statement. Please comment and revise.

Author Response

Answer to Reviewer 1

Dear author and editor. I find the paper interesting, relevant and well performed and should be published. I have a few comments that I think may be relevant:

We thank the reviewer for attention to our manuscript as well as for valuable comments and recommendations. The answers to the comments are presented below.

Q1. The quality/resolution of figures 1 and 2 needs to be improved.

A1. Agree. The quality has been improved.

Q2. The text in figure 3 is too small for old eyes.

A2. The corresponding correction has been made.

Q3. On page 6 it is claimed that the slopes in figure 3 are all 45 degrees. It does not look like that in the figure. Please comment or revise. 

A3. Thank you or your remark! The text has been revised accordingly.

Q4. On page 11 it says that the atomic % ratios are homogeneous and near the target value. Figure 8d clearly disagrees with this statement. Please comment and revise.

A4. Thank you very much for your comment. We have modified Figure 8 and selected a better axis to represent the distribution of atoms in the active particle. We have also removed the data on distribution of carbon for better clarity. From the presented graph (Fig. 8d) the conclusion about the uniform character of distribution is easier to draw.

Figure 8. Data of EDX mapping for the single Ni-Cu particle of the Ni-Cu-CNF composite obtained via the decomposition of ethylene over the NiCu(3) sample at 550 °C for 1 min: (a) distribution of carbon, nickel, and copper; (b) distribution of copper; (c) distribution of nickel; (d) atomic fraction of the elements along the particle length (shown by the yellow line).

Reviewer 2 Report

The current study describes synthesis of Ni-Cu-CNF Composite Materials via carbon erosion of Ni-Cu bulk form was previously prepared by mechanochemical technique. The interesting of the current study is average and the authors discussed this method with the same composition in their previous works. I found the topic is very interesting due to the wide range of the application depending on the CNF Composite Materials.

The manuscript is well written and described the outcome data very well. Thus, I recommend accepting the manuscript after fixed these minors issues:

1-     The authors should add in the introduction part their attempts to solve the proplem they suggested in the introduction, where the authors said “However, some questions regarding the CCVD process remained unclear. The features of CCVD of ethylene over the microdispersed Ni- Cu alloys, as well as the effects of the pretreatment conditions, were not studied precisely yet.” What about these studies that already cited in the current version of the manuscript;

https://link.springer.com/article/10.1134/S0023158422010049, https://link.springer.com/article/10.1007/s11244-022-01739-7,

https://www.mdpi.com/1996-1944/15/24/9033?type=check_update&version=2

The authors should explain this in details in the revised version of the manuscript to illustrate the novelty of the current work.

2-     The resolution of Figure 1 is very low, I hardly seen the details of it, it will be graet if the authors add better resolution in the revised version of the manuscript.

3-     For the Synthesis of the Ni-Cu-CNF Composites why the authors heated the reactor up to the target temperature of 550 °C, why this specific temperature? What about higher or lower than this temperature? The authors should explain that in the revised version of the manuscript.   

4-     The authors should add the uncertainties of the data in Table 1, Table 2 and Table 3.

5-     The author should add more information about preparing the samples for TEM analysis. If possible, the authors add FFT and SAED pattern to Figure 10.

Author Response

Answer to Reviewer 2

The current study describes synthesis of Ni-Cu-CNF Composite Materials via carbon erosion of Ni-Cu bulk form was previously prepared by mechanochemical technique. The interesting of the current study is average and the authors discussed this method with the same composition in their previous works. I found the topic is very interesting due to the wide range of the application depending on the CNF Composite Materials.

The manuscript is well written and described the outcome data very well. Thus, I recommend accepting the manuscript after fixed these minors issues:

We thank the reviewer for attention to our manuscript as well as for valuable comments and recommendations. The answers to the comments are presented below.

Q1. The authors should add in the introduction part their attempts to solve the problem they suggested in the introduction, where the authors said “However, some questions regarding the CCVD process remained unclear. The features of CCVD of ethylene over the microdispersed Ni- Cu alloys, as well as the effects of the pretreatment conditions, were not studied precisely yet.” What about these studies that already cited in the current version of the manuscript;

https://link.springer.com/article/10.1134/S0023158422010049, https://link.springer.com/article/10.1007/s11244-022-01739-7,

https://www.mdpi.com/1996-1944/15/24/9033?type=check_update&version=2

The authors should explain this in details in the revised version of the manuscript to illustrate the novelty of the current work.

A1. Thank you for your comment. Indeed, this work is a continuation of the study of the Ni-Cu system that we have previously described. However, it should be noted that in this case we have applied the TPR-H2 method for a more detailed analysis of the phase composition of the alloy. Also, a detailed examination (XRD, electron microscopy) was carried out to study the evolution of the self-dispersing catalytic system on each stage, including mechanochemical activation, heating to reaction temperature and subsequent disintegration of Ni-Cu alloy with formation of active particles during catalytic pyrolysis. In addition, the high resolution TEM method combined with EDX-mapping was used in this study to visualize the first step of the carbon erosion process and to examine the uniformity of distribution for Ni and Cu elements throughout the active particles catalyzing the growth of carbon filaments.

It is also worth emphasizing that the purpose of the current study was to reveal the regularities for creation of a composite metal-carbon material. In this regard, the paper also contains novel results on the description of the synthesized composites (SEM, TEM, and BET data).

The corresponding information has been added to the Introduction of manuscript.

Q2. The resolution of Figure 1 is very low, I hardly seen the details of it, it will be graet if the authors add better resolution in the revised version of the manuscript.

A2. Thank you! The improvement has been done.

Q3. For the Synthesis of the Ni-Cu-CNF Composites why the authors heated the reactor up to the target temperature of 550°C, why this specific temperature? What about higher or lower than this temperature? The authors should explain that in the revised version of the manuscript. 

A3. Thank you for your suggestion! According to the results of recently conducted experiments, the target temperature of 550°C was selected as basic since it provides stable operation of catalysts in the ethylene decomposition reaction [1], which makes it possible to calculate various kinetic parameters. Also, it turned out to be optimal in the ethylene catalytic pyrolysis reaction performed on the laboratory tubular reactor. This mentioning has been added to the text.

It should be noted that the described studies of the effect of reaction temperature on catalytic activity of Ni-Cu samples are being prepared for publication. We believe that the addition of this information to the revised version of paper would make it overloaded with information; therefore we decided to prepare the separate publication on this topic.

[1]. Afonnikova S.D., Mishakov I.V., Bauman Y.I., Trenikhin M.V., Shubin Y.V., Serkova A.N., Vedyagin A.A. // Topics in Catalysis. 2023. V.66. P.393–404. DOI: 10.1007/s11244-022-01739-7.

Q4. The authors should add the uncertainties of the data in Table 1, Table 2 and Table 3.

A4. Thank you for your comment! The corresponding information has been added to Tables 1-3, as well as to Experimental section.

Q5. The author should add more information about preparing the samples for TEM analysis. If possible, the authors add FFT and SAED pattern to Figure 10.

A5. The corresponding correction has been done. Unfortunately, we cannot provide data on FFT and SAED analysis, but we will take your suggestion into account in future studies.

Round 2

Reviewer 1 Report

Dear author and editor

With the corrections made, I find the paper ready for publishing.

with kind regards

Reviewer 2 Report

The authors answer correctly with all my comments. The manuscript can publish in a present form.